# Hybrid Azine Derivatives: A Useful Approach for Antimicrobial Therapy

**DOI:** 10.3390/pharmaceutics14102026

**Published:** 2022-09-23

**Authors:** Dorina Amariucai-Mantu, Violeta Mangalagiu, Iustinian Bejan, Aculina Aricu, Ionel I. Mangalagiu

**Affiliations:** 1Faculty of Chemistry, Alexandru Ioan Cuza University of Iasi, 11 Carol 1st Bvd, 700506 Iasi, Romania; 2CERNESIM Centre, Institute of Interdisciplinary Research, Alexandru Ioan Cuza University of Iasi, 11 Carol I, 700506 Iasi, Romania; 3Chemistry of Natural and Biologically Active Compounds Laboratory, Institute of Chemistry, 3 Academiei Str., MD-2028 Chisinau, Moldova

**Keywords:** hybrid compounds, antimicrobial, pyridine, quinoline, isoquinoline, fused azine

## Abstract

Nowadays, infectious diseases caused by microorganisms are a major threat to human health, mostly because of drug resistance, multi-drug resistance and extensive-drug-resistance phenomena to microbial pathogens. During the last few years, obtaining hybrid azaheterocyclic drugs represents a powerful and attractive approach in modern antimicrobial therapy with very promising results including overcoming microbial drug resistance. The emphasis of this review is to notify the scientific community about the latest recent advances from the last five years in the field of hybrid azine derivatives with antimicrobial activity. The review is divided according to the main series of six-member ring azaheterocycles with one nitrogen atom and their fused analogs. In each case, the main essential data concerning synthesis and antimicrobial activity are presented.

## 1. Introduction

According to the WHO, infectious diseases caused by microorganisms represent a major threat that affects society and human health, exerting great pressure on health systems, individuals and communities [1]. In particular, overconsumption and widespread use and misuse of antimicrobial agents have resulted in the emergence of drug resistance, multi-drug resistance and extensive-drug-resistance phenomena to microbial pathogens and many other drawbacks (toxicity and non specificity of drugs, high prices, etc.). So far, searching for new chemical entities with improved antimicrobial properties remains a very challenging and important task in medicinal chemistry.

During the last few years, molecular hybridization represents a powerful tool in drug design, by merging two or more drug pharmacophores in a single hybrid multi-functional molecule. Usually, the resulting hybrid entity has superior properties compared with conventional classic drugs, with dual or multiple target mechanisms, better biological activity and specificity, less side effects and toxicity, less drug–drug interactions, etc. [2,3]. As a result of this approach, important advances have been achieved in antimicrobial therapy, some of the present drugs from the market have a hybrid structure (Figure 1) and some hybrid structures are in different clinical trials (Figure 2) [2,3,4,5,6,7,8].

A literature survey revealed that azines are privileged scaffolds in current medicinal chemistry and drug discovery, possessing a large variety of biological activities, such as: antibacterial, antifungal, antiplasmodial and antimalarial, anthelmintic, antitubercular, antiviral, anticancer, anti-inflammatory, antihypertensive, diuretic, antithrombic, anticoagulant, antidepressant, anxiolytic, anticonvulsant, analgesic, antiulcer, antidiabetic, antihistaminic, etc. [4,5,6,7,8]. As a matter of fact, the greatest majority of the existing drugs from the market contain in their structure a nitrogen heterocycle, some of them being a hybrid structure (Figure 1), which justifies the demand of the pharmaceutical industry for such drugs with nitrogen heterocycle skeleton.

Because of the above considerations, there is a large and urgent demand from the pharmaceutical industry for newer and better drugs with enhanced antimicrobial activity, with superior pharmacokinetic and pharmacodynamic properties, the hybrid drugs being a serious and preferential option.

In this review, we present an overview of the newest research concerning the synthesis and antimicrobial activity of hybrid azine derivatives. The main data reviewed in this paper are summarized in Table 1 presented below.

## 2. Results and Discussion

### 2.1. Six-Member Ring Azaheterocycles with One Nitrogen Atom. Hybrid Pyridine

In their attempt to identify new antimicrobial compounds, Eryılmaz et al. [9] designed and synthesized different hybrid pyridine derivatives bearing in the 2- and 4-position of the ring of a thiazole moiety. The synthesis was straight and efficient, involving a Hantzsch cyclocondensation of pyridine-2- and 4- carbothioamide **1** and **3** with acetophenone derivatives, when the desired hybrid 4-(*R*-2-yl)-2-(pyridin-2-yl)thiazole **2a**–**e** and 4-(*R*-2-yl)-2-(pyridin-4-yl)thiazole **4a**–**e** are obtained, Figure 1. The synthesized compounds were tested for their antibacterial activity [four strains, *Gram-positive* (*Bacillus cereus*, *Staphylococcus aureus*) and *Gram-negative* (*Escherichia coli*, *Pseudomonas aeruginosa*)] and antifungal activity (one strain, *Candida albicans*) *via* minimal inhibitory concentration (MIC) method and DNA cleavage activity studies. The authors established interesting correlation structure-biological activity (SAR), the most relevant finding being that 4-pyridine thiazole hybrid compounds **4a**–**e** showed more potent activity than **2a**–**e**. The most promising compound was found to be **4c** (MIC values 0.01 mM) exhibited on the bacterial strains *Staphylococcus aureus* and *Bacillus cereus*.

In a subsequent paper, some of the above authors (Cinarli et al. [10]) synthesized different hybrid aroylhydrazone-pyridine-metal derivatives. The newly hybrid aroylhydrazone-pyridine metal derivatives [ZnL_2_] **7** have been synthesized in two steps: an initial cyclocondensation of pyridine-2-acyl derivative **5** (with aroylhydrazone leading to pyridine-aroylhydrazone ligand **6**) is followed by complexation with M^2+^ metal (Zn^2+^), Figure 2.

The synthesized compounds were tested for their antibacterial activity (four strains, *Pseudomonas aeruginosa*, *Escherichia coli*, *Bacillus cereus* and *Staphylococcus aureus*) and antifungal (one strain, *Candida albicans*) activity *via* minimal inhibitory concentration method. The [ZnL_2_] **7** has been found to be more active than pyridine-aroylhydrazone ligand **6** in all microorganisms (MIC = 11.71 μg/mL for bacteria and MIC = 23.43 μg/mL for *C. albicans*). The authors claim that the synthesized new complex acts on microorganisms by disrupting the cell wall structure. The DNA binding interactions was also determined experimentally by spectrophotometric and electrochemical methods. The obtaining data indicate that ligand **6** and hybrid [ZnL_2_] **7** interact the most with guanine base, and charge transfer is from DNA guanine bases to the molecular structures. Moreover, antioxidant activity was determined, and the hybrid [ZnL_2_] **7** acted as a scavenger against peroxide radicals.

Trotsko et al. [11] designed and synthesized different hybrid pyridine derivatives bearing at the 2-, 3- or 4- position of the ring of a thiazolidine-2,4-dione moiety. The synthesis involve a condensation reaction of hydrazonyl-pyridine **8a**–**c** with the corresponding (2,4-dioxo-1,3-thiazolidin-5-yl/ylidene) **9a**,**b**/**10a**–**c**, which are leading to the desired hybrid pyridine-2,4-dioxo-1,3-thiazolidin-5-yl derivatives **11a**–**f** or pyridine-2,4-dioxo-1,3-thiazolidin-5-ylidene derivatives **12a**–**i**, Figure 3.

The in vitro antimycobacterial assay (*Mycobacterium tuberculosis*) of the newly obtained compounds reveals strong activity in the concentration range of 1–512 μg/mL and low cytotoxicity. Interesting SAR correlations have been performed, and the highest antimycobacterial activity (MIC = 1 μg/mL) was demonstrated for the hybrid pyridine derivatives bearing the thiazolidine-2,4-dione moiety at the 4-position of the pyridine ring (hybrids **11a**–**c** and **12g**–**i**).

Sanad et al. [12] have performed an interesting study concerning the in vitro antimicrobial activity of some newly hybrid thieno-pyrimidin-pyridine derivatives. The synthesized compounds belonged to different classes of substituted pyridine: thiophen-dihydropyridine **14**, thiophen-pyrido-pyrimidin-4(1*H*)-one **15**, and fused pyridine: pyrido-thiophen-triazolo-pyrimidine **16a**–**c**, thiophen-pyrido-thieno derivative **17**, thiophen-pyrido-thieno-pyrimidin-4-one **18**, thiophen-pyrido-thieno-pyrimidin-2,4-dione **19**, thiophen-pyrido-thieno-pyrimidin-2-*R*-4-one **20**, Figure 4.

The synthetic approach is straight and efficient, involving typical organic chemistry reactions, mostly cyclocondensations. The synthesized compounds were tested in vitro for their antibacterial activity against *Escherichia coli* and *Klebsiella pneumoniae* as *Gram-negative* bacterial strains as well as against *Staphylococcus aureus* and *Streptococcus*
*mutans* as *Gram-positive* bacterial strains. The obtained results (expressed as the diameter of inhibition zones (DIZ) and MIC) reveal that the thiophen-pyrido-thieno-pyrimidin-2-*R*-4-one **20a**,**b** exhibit the strongest antibacterial activities against all the tested bacteria, in the range of 40–60 mm for inhibition zones, respectively, 4–16 μg/mL for MIC values.

Desai et al. [13] have studied the in vitro antimicrobial activities of some newly hybrid oxazino-pyridine derivatives. The desired compounds, oxazin-3(4*H*)-yl)phenyl)ethyildene)amino)-6-((arylidene)amino)-4-(4-chlorophenyl)-2-oxo-1,2-dihydropyridine **23a**–**j**, were synthesized in two steps, by cyclocondensation of oxazine **21** followed by condensation of the intermediate **22**, Figure 5.

The synthesized hybrid compounds were tested for their in vitro antibacterial activity against various bacteria (*Escherichia coli*, *Pseudomonas aeruginosa*, *Staphylococcus aureus*, *Streptococcus pyogenes*) and fungus (*Candida albicans*, *Aspergillus niger*, *Aspergillus clavatus*) *via* the MIC method. Some compounds have proved to have a very powerful activity against bacteria *E. coli* (**23h**, MIC = 25 μg/mL) and against fungus *C. albicans* (**23f**, MIC = 50 μg/mL), respectively, *A. clavatus* (**23h**, MIC = 25 μg/mL).

Sribalan et al. [14] have studied thein vitroantimicrobial activity of some tetrazole-heterocycle hybrid derivatives. The synthesis supposes a cyclocondensation reaction of amide precursors **24** with sodium azide, when the corresponding tetrazolo-pyridine **25a**–**d** and tetrazolo-quinoline **26a**–**e** hybrids are obtained, Figure 6.

The synthesized tetrazolo-pyridine **25a**–**d** and tetrazolo-quinoline **26a**–**e** hybrids were tested for their in vitro antibacterial activity against various bacteria (*Klebsiella pneumoniae*, *Pseudomonas aeruginosa*, *Staphylococcus aureus*, *Streptococcus pyogenes*) and the fungus *Candida albicans*. An interesting SAR correlation has been performed. The compound **25a** (the pyridyl ring is decorated with *n*-butyl) proved to be the most active from the tetrazolo-pyridine series against all bacteria (DIZ in the range of 4–15 mm), having a superior inhibition to the standard drug (amikacin). The compound **26d** (the quinoline ring is decorated with a piperidyl-sulfonamide moiety) proved to be the most active from the tetrazolo-quinoline series against all bacteria (DIZ in the range of 4–10 mm), having a comparable inhibition to the standard. The antifungal activity was negligible.

Kuthyala et al. [15] have studied the in vitro antimicrobial activity of some oxadiazolo-imidazopyridine hybrid derivatives. The synthesis was straight, involving a cyclocondensation reaction of hydrazonyl-imidazopyridine **27** with different benzoic acids, when the corresponding oxadiazolo-imidazopyridine hybrids **28a**–**j** were obtained, Figure 7.

The synthesized oxadiazolo-imidazopyridine hybrids **28a**–**j** were tested for their in vitro antibacterial activity against various human bacterial pathogens (*Escherichia coli*, *Klebsiella pneumoniae*, *Staphylococcus aureus*, *Bacillus subtilis*) and the fungus *Candida albicans* and *Aspergillus niger*. An interesting SAR correlation has been performed. The compounds **28f** and **28g** have high activity against *Gram-positive* bacteria *S. aureus* (MIC = 3.12 μg/mL), while compound **28f** proved to have high activity against fungus *C. albicans* (MIC = 12.5 μg/mL).

Ahirwaret al. [16] synthesized two new series of some 1,3,4-triazolo-pyridine hybrid derivatives and studied their antimicrobial activities. The synthesis was conducted in two steps: a cyclocondensation reaction of dithiocarbazate **29** with ammonia leading to the first class of hybrids triazolo-pyridine **30a**–**n**, then an alkylation reaction of **30a**–**n** with benzyl halide takes place leading to the second class of hybrids triazolo-pyridine **31a**–**n**, Figure 8.

The synthesized triazolo-pyridine hybrids **30a**–**n** and **31a**–**n** were evaluated for their in vitro antibacterial activity against *Gram-positive* bacteria (three strains: *Staphylococcus aureus*, *Streptococcus pyogenes*, *Enterococcus faecalis*) and *Gram-negative* bacteria (three strains: *Escherichia coli*, *Pseudomonas aeruginosa*, *Acinetbacter baumannii*) by MIC assay. From the tested compounds, two of them, **31h** and **31i**, have excellent activity against all strains (MIC in the range of 0.91–11 μg/mL).

Jaabil et al. [17] have studied the in vitro antimicrobial activities of some newly hybrid 1,2,3-triazolo-pyridine derivatives. The synthesis was green and efficient, under grinding strategy at room temperature, involving *one-pot* sequential multicomponent reactions of aryl aldehydes **32a**–**r**, malonitrile **33**, methanol and 1,2,3-triazolyl ketone **34**, when the corresponding 1,2,3-triazolyl-pyridine/cyanopyridine hybrids **35a**–**r** were obtained, Figure 9.

The synthesized 1,2,3-triazolo-pyridine hybrids **35a**–**r** were screened for their in vitro antibacterial activity against three human bacterial strains, *Staphylococcus aureus*, *Salmonella typhi* and *Escherichia coli*, using the MIC method. Some of the 1,2,3-triazolyl cyanopyridine hybrids displayed a remarkable activity against the tested germs, better than tetracycline (standard drug), according to the R-substituent from the phenyl ring. The most active compounds were **35c** (with R = −4-chloro-; MIC in the range of 50–90 μg/mL), **35e** (with R = −2-methyl-; MIC in the range of 40–90 μg/mL) and **35r** (with R = −2-thienyl; MIC in the range of 70–120 μg/mL). The hybrid 1,2,3-triazolo-pyridine compounds were also tested for their antioxidant activity in the assay by 2,2-diphenyl-1-picrylhydrazyl (DPPH) method, showing promising results.

Felefel et al. [18] synthesized three new series of some pyridine hybrid derivatives (namely pyrazole-pyridine **37**–**41**, triazolo-pyridine **42**–**45** and triazino-pyridine **46**) and studied their antimicrobial activities. The synthesis is using as starting material 6-(3,4-dimethylphenyl)-2-hydrazinyl-4-(thiophen-2-yl)-pyridine-3-carbonitrile **36** which react with different compounds with methylene active group (namely acetyl acetone, diethylmalonate, ethyl cyanoacetate, ethyl benzoylacetate and/or ethyl acetoacetate) to produce the desired pyrazole-pyridine hybrid derivatives **37**–**41**, Figure 10.

The synthesis of triazolo-pyridines **42**–**45** and tetrazolo-pyridines **46** use as starting material the same intermediate, the 6-(3,4-dimethylphenyl)-2-hydrazinyl-4-(thiophen-2-yl)-pyridine-3-carbonitrile **36**, which react with the appropriate formic acid, acetic acid, benzoyl chloride, carbon disulfide, respectively, sodium nitrite, to produce the desired hybrid derivatives **42**–**45** and **46**, Figure 11.

The synthesized pyridine hybrids **37**–**46** were screened for their in vitro antibacterial activity against *Gram-positive* bacteria (*Staphylococcus aureus* and *Bacillus subtilis*), *Gram-negative* bacteria (*Salmonella typhi* and *Escherichia coli*) and fungus (*Aspergillus flavus* and *Candida albicans*) using the disk diffusion agar technique. Some of the hybrids have significant antimicrobial activity, the most active compounds being **37** with a DIZ in the range of 10–17 mm. The antioxidant activity was also tested.

Amperayani et al. [19] synthesized a library of piperine-pyridine hybrid derivatives and studied their antimicrobial activities. The reaction pathway is straight, in one step, involving an acylation reaction of various amino-pyridine derivatives **47a**–**h**, when the corresponding hybrids piperine-pyridine derivatives **48a**–**h** are obtained, Figure 12.

The synthesized piperine-pyridine hybrid derivatives **48a**–**h** were tested for their in vitro antibacterial activity against some *Gram-positive* and *Gram-negative* bacterial strains (*Bacillus subtilis*, *Streptobacillus*, *Staphylococcus aureus*, *Escherichia coli*, *Klebsiella pneumoniae*, *Pseudomonas aeruginosa*, *Enterococcus faecalis* and *Salmonella typhi*) and fungus strains (*Aspergillus niger*, *Aspergillus flavus*, *Aspergillus fumigatus* and *Candida albicans*) using the disk diffusion agar technique. The piperine-pyridine hybrids **48a**, **48d** and **48h** have very good activity against the *Gram-negative* strains *E. coli*, *K. pneumoniae*, *E. faecalis* and *P. aeruginosa*, having a DIZ in the range of 22–26 mm, superior to control standard drug). The antifungal activity of hybrids was moderate.

### 2.2. Six-Member Ring Azaheterocycles with One Nitrogen Atom. Hybrid Quinoline and Isoquinoline

In their attempt to obtain new quinoline derivatives with antimicrobial activity, Albayrak et al. [20] synthesized a library of 20 new triazolo-quinoline hybrid derivatives and studied their antimicrobial activities. The reaction pathway involves several steps (Figure 13), starting from 8-nitroquinoline **53**. The initial reduction reaction of **53** is leading to 8-aminoquinoline **54**, which is suffering a subsequent *N*-alkylation with azido-iodo-propane **52a**,**b** (generated from the corresponding bromo-alkyl alcohol) leading to alkyl-azide-quinolines **55** and **56**. Finally, the alkyl-azide-quinoline derivatives are treated with the corresponding alkyne **57a**–**j** leading to the desired products, the triazolo-quinoline hybrid derivatives **58a**–**j** and **59a**–**j**.

The synthesized triazolo-quinoline hybrid derivatives **58a**–**j** and **59a**–**j** were tested for their in vitro antibacterial activity against some *Gram-positive* and *Gram-negative* bacterial strains (*Bacillus subtilis*, *Streptococcus pneumoniae*, *Staphylococcus aureus*, *Escherichia coli*, *Klebsiella pneumoniae*, *Pseudomonas aeruginosa* and *Enterococcus faecalis*) and fungus strains (*Candida parapsilosis* and *Candida albicans*) using the disk diffusion agar technique. The triazolo-quinoline hybrid derivatives **58a**–**j** and **59a**–**j** manifest good activity against the tested strains. The most active compound was **58a**, having excellent activity against *E. coli*, *P. aeruginosa*, *K. pneumoniae*, *E. faecalis*, *S. aureus*, *S. pneumoniae*, *B. subtilis*, *C. albicans* and *C. parapsilosis*. In some cases, the activity was several orders of magnitude superior to control drugs (DIZ of **58a** was in the range of 35–250 mm; control, ampicillin, respectively, fluconazole have had a DIZ of 35 mm).

Hryhoriv et al. [21,22] synthesized two new classes of hybrid derivatives analogous to fluoroquinolones, namely piperidino-quinoline **61a**,**b** and 1,2,3-triazolo-piperidino- quinoline **62a**–**k**, and studied their antimicrobial activities. The first class of hybrids was obtainedviaan *N*-alkylation reaction of piperidino-quinoline **60a**,**b,** when the *N*-substituted-piperidino-quinoline hybrids **61a**,**b** are obtained. A click cyclocondensation reaction of **61a**,**b** occurs to the second class of hybrids, the 1,2,3-triazolo-piperidino-quinoline **62a**–**k**, Figure 14.

The synthesized hybrid derivatives piperidino-quinoline **61a**,**b** and 1,2,3-triazolo-piperidino-quinoline **62a**–**k** were tested for their in vitro antibacterial activity against standard bacterial strains *Staphylococcus aureus* and *Escherichia coli*, respectively, and the fungus *Candida albicans* using the disk diffusion agar technique. The antimicrobial assay was also made by some clinical bacterial strains *S. aureus* and *E. coli*, respectively, and fungus *C. albicans* using the same method. The hybrid, 1,2,3-triazolo-piperidino-quinoline **62c** have a very good activity against the tested standard strains (DIZ in the range of 25–35 mm), having a superior inhibition zone to control (DIZ = 25 mm). Against clinical microbial strains, the activity was negligible.

Drweesh et al. [23] synthesized hybrid organic-inorganic derivatives and studied their antimicrobial activities, antiproliferative activity, and radical scavenging properties. In order to synthesize the desired palladium-quinoline derivatives **64a**–**d**, they used organic cation modulation, doing a complexation reaction with PdCl_2_ of the quinolines **63a**–**d**, Figure 15.

The synthesized palladium-quinoline derivatives hybrids **64a**–**d** and the free ligands **63a**–**d**, were tested for their in vitro antimicrobial activity against 14 standard microbial strains (*Gram-positive* and *Gram-negative* bacteria, fungus: *Bifidobacterium animalis*, *Lactobacillus plantarum*, *Bacillus subtilis*, *Staphylococcus aureus ATCC 663*, *Staphylococcus aureus ATCC 25923*, *Pseudomonas aeruginosa*, *Proteus mirabilis *, *Escherichia coli*, *Salmonella enterica*, *Candida albicans*, *Saccharomyces boulardii*, *Aspergillus flavus*, *Trichoderma viridae*, *Aspergillus niger*). All hybrid compounds **64a**–**d** have high antimicrobial activity against all tested strains, with minimum inhibitory concentration values ranging from 1.95 to 250 μg/mL. The results of DNA interaction studies indicate that the hybrids **64a**–**d** and the free ligands **63a**–**d**, interact with the DNAvia an intercalation mechanism (the aromatic chromophore intercalates the base pairs of DNA; compound **64a** has the highest binding affinity). The anticancer activity was also studied, with compounds **64a** and **64b** having selective and high cytotoxicity against human lung and breast cancer cells.

Nehra et al. [24] synthesized a series of triazole-benzothiazole-quinoline hybrids and studied their antimicrobial properties. The reaction pathway is straight and efficient (Figure 16), involving a click cyclocondensation reaction of azido-alkyl-benzothiazole **65a**–**f** (generated in situ from the corresponding bromo-alkyl derivative) with the corresponding alkyne-quinoline, leading to the desired products, triazole- -benzothiazole-quinoline hybrids **66a**–**f**.

The synthesized hybrids **66a**–**f** were evaluated for their in vitro antimicrobial activity against two *Gram-positive* strains (*Staphylococcus aureus* and *Bacillus subtilis*) and two *Gram-negative* strains (*Escherichia coli* and *Pseudomonas aeruginosa*) and two fungal strains (*Candida tropicalis* and *Aspergillus terreus*). The tested hybrids have good antimicrobial activity against both bacteria and fungus. The most promising compound was proved to be **66a**, with an antibacterial (DIZ in the range of 15–17 mm) and antifungal (DIZ in the range of 21–34 mm) activity superior to reference ciprofloxacin (DIZ = 22 mm) and fluconazole (DIZ = 20 mm), respectively. Interesting molecular docking studies were also performed.

Awolade et al. [25] synthesized a library of triazole-quinoline hybrids and studied their antimicrobial properties. The reaction pathway is straight involving click chemistry of various azides with triple bond derivatives, *via* copper(I)-catalyzed azide-alkyne 3 + 2 dipolar cycloaddition reactions, Figure 17.

The synthesized hybrids **67a**–**u**, **68a**–**z**, **69a**–**n** and **70a**,**b** were evaluated for their in vitro antimicrobial activity against ESKAPE microbial strains (bacteria and fungus: (*Staphylococcus aureus*, *Escherichia coli*, *Acinetbacter baumannii*, *Klebsiella pneumoniae*, *Candida albicans* and *Candida neoformans*). Some of the compounds proved to have a good and broad-spectrum of antibacterial activity, against methicillin-resistant *S. aureus* (MRSA), *E. coli*, *A. baumannii*, multidrug-resistant *K. pneumoniae* and the fungus *C. albicans* and *C. neoformans* (superior to control, fluconazole). The most promising antibacterial compound was proved to be **70b** with an MIC = 75.39 μM against MRSA, *E. coli*, *A. baumannii*, and multidrug-resistant *K. pneumoniae*. The hybrid **70b** also has a very good antifungal activity against *C. albicans* and *C. neoformans* with an MIC of 37.69 and 2.36 μM, respectively, superior to control fluconazole.

Ammar et al. [26] synthesized a series of thiazole-quinoline hybrids and studied their antimicrobial properties. In order to synthesize the desired compounds, they used the condensation reaction between formil-quinoline derivatives with amino-thiazole or sulfathiazole, when the desired Schiff’s base thiazole-quinoline **71** and **72**, are obtained, Figure 18.

Further, the condensation reaction between formil-quinoline derivatives with different thiazolone derivatives lead to hybrid thiazolone-quinoline derivatives **73**–**76**, Figure 19.

Finally, the cyclization of different quinoline-thiosemicarbazone derivatives with the halogenated compounds lead to other hybrid thiazole-quinoline derivatives **77**–**82**, Figure 20.

The synthesized hybrids **71**–**82**, were evaluated for theirin vitroantimicrobial activity against eight standard microbial strains, three *Gram-positive* bacteria (*Staphylococcus aureus*, *Bacillus faecalis* and *Bacillus subtilis*), three *Gram-negative* bacteria (*Escherichia coli*, *Salmonella typhi* and *Pseudomonas aeruginosa*), and two fungi (*Candida albicans* and *Fusarium oxysporum*). Some of the compounds have good antimicrobial activity, with MIC and MBC values ranging between 0.95 and 62.5 µg/mL, and 1.94 and 118.7 µg/mL, respectively. Two compounds, namely **77b** and **73a**, proved to be the most active of the series against *S. aureus* and *E. coli* having an MIC between 0.95 and 7.81 μg/mL, respectively a MBC between 3.31 and 15.62 μg/mL.

Using a similar strategy, some of the above authors (Eissa et al. [27]) synthesized a new series of thiazole-quinoline hybrids and studied their antimicrobial properties. In order to synthesize the desired compounds, they used the cyclization of quinoline-thiosemicarbazone derivatives with the halogenated compounds, when the corresponding hybrid thiazole-quinoline derivatives, **83a**–**f**, **84a**–**f** and **85a**–**f** are obtained, Figure 21.

The synthesized hybrids **83a**–**f**, **84a**–**f** and **85a**–**f**, were evaluated for their in vitro antimicrobial activity against *Gram-positive* (five strains: *Staphylococcus aureus*, *Staphylococcus epidermidis*, *Streptococcus pyogenes*, *Bacillus subtilis* and *Enterococcus faecalis*) and *Gram-negative* bacteria (five strains: *Neisseria gonorrhoeae*, *Proteus vulgaris*, *Klebsiella pneumonia*, *Shigella flexneri* and *Pseudomonas aeruginosa*), as well as fungus (five strains: *Aspergillus fumigatus*, *Aspergillus clavatus*, *Candida albicans*, *Geotrichum candidum*, and *Penicillium marneffei*). Some of the compounds displayed good antimicrobial activity, superior to the used control. The most active compound was found to be **85e**, having a two-fold potency compared with gentamycin for inhibition of *N. gonorrhoeae*, four-fold potency compared with amphotericin B for the inhibition of *A. fumigatus*, equipotent activity compared with the reference drugs for inhibition of *S. flexneri*, *S. pyogenes*, *P. vulgaris*, *A. clavatus*, *G. candidum* and *P. marneffei*.

Lagdhir et al. [28] synthesized a library of piperazin-quinoline hybrids and studied their antimicrobial properties. The reaction pathway involves two steps (an alkylation and a condensation reaction), leading to the piperazin-quinoline hybrids **86a**–**l**, Figure 22.

The synthesized hybrids **86a**–**l** were evaluated for their in vitro antibacterial (*Staphylococcus aureus*, *Streptococcus pyogenes*, *Escherichia coli* and *Pseudomonas aeruginosa*) and antifungal (*Aspergillus clavatus*, *Aspergillus niger* and *Candida albicans*) activity, antimalarial (*Plasmodium falciparum*) and antituberculosis (*Mycobacterium tuberculosis*) activity. Some of the compounds have good antibacterial and antifungal activity against *S. aureus* and *C. albicans*. The hybrids **86a**, **86b**, **86d**, **86j** and **86k**, are the most active as an antimicrobial against *S. aureus*, having an MIC = 100 μg/mL, equal to the control drug ampicillin. The hybrid **86k** has excellent antifungal activity against *C. albicans*, having an MIC = 250 μg/mL, two folds higher compared with the control drug griseofulvin. The antimalarial and antitubercular activity proved to be moderate for the majority of compounds.

Desai et al. [29] synthesized a series of pyridine-quinoline hybrids and evaluated it for their antimicrobial properties. The reaction pathway involves a cyclocondensation reaction of quinoline derivative with benzylidene-malononitril, when the corresponding pyridine-quinoline hybrids **87a**–**j** were obtained, Figure 23.

The synthesized hybrids **87a**–**j** were evaluated for their in vitro antimicrobial activity against *Gram-positive* (two strains: *Staphylococcus aureus* and *Staphylococcus pyogenes*) and *Gram-negative* (two strains: *Escherichia coli* and *Pseudomonas aeruginosa*) bacteria, as well as to fungus (three strains: *Aspergillus clavatus*, *Aspergillus niger* and *Candida albicans*). Some of the compounds displayed promising antimicrobial activity. The hybrid **87i** has the best antibacterial activity against *E. coli*, *P. aeruginosa* and *S. aureus* strains, with an MIC = 12.5 μg/mL, two folds higher compared with the control drug ciprofloxacin (MIC = 25 μg/mL). The most active compound against *C. albicans* was found to be **87e**, having an MIC=25 μg/mL, much better compared with the control drug griseofulvin (MIC = 500 μg/mL).

Vishnuvardhan et al. [30] synthesized a library of triazole-quinoline hybrids and studied their antimicrobial properties. The reaction pathway involves a typical click cyclocondensation reaction of quinoline with a triple bond with aryl-azide derivatives, when the corresponding triazole-quinoline hybrids **88a**–**l**, Figure 24.

The synthesized hybrids **88a**–**l** were evaluated for theirin vitroantimicrobial activity against *Gram-positive* (*Staphylococcus aureus* and *Enterococcus faecalis*) and *Gram-negative* (*Escherichia coli* and *Pseudomonas aeruginosa*) bacteria, as well as to fungus (*Aspergillus niger* and *Candida albicans*). Most of the hybrid compounds have good antimicrobial activity. The best antibacterial activity reveals the hybrids **88d**, **88h** and **88i**, having a DIZ in the range of 16–21 mm, superior to control ampicillin (DIZ = 15 mm). The best antifungal activity reveals the hybrids **88d**, **88h** and **88k**, having a DIZ in the range of 18–27 mm, superior to control griseofulvin (DIZ = 17 mm).

Abdel-Rahman et al. [31] synthesized a series of piperazin-quinoline hybrids derived from ciprofloxacin and studied their antimicrobial and anticancer properties. The reaction pathway involves the reaction of ciprofloxacin with the corresponding phenolic derivatives with an excess of formaldehyde, when the piperazin-quinoline hybrids **89a**–**j** are obtained, Figure 25.

The synthesized hybrids **89a**–**j** were evaluated for their antimicrobial and anticancer activity. The antibacterial screening was preconformed on *Gram-positive* and *Gram-negative* strains: *Staphylococcus aureus*, MRSA clinical strain, MRSA reference strain, *Escherichia coli* and *Pseudomonas aeruginosa*. The obtained results reveal that the hybrid **89d** has the best antibacterial activity against *S. aureus*, MRSA (reference strain) and MRSA (clinical strain) with an MIC of 0.57, 0.52, and 0.082 µg/mL, respectively, (compared with the reference standard drug ciprofloxacin which has an MIC of 1.63 µg/mL against *S. aureus*, an MIC of 1.45 µg/mL against MRSA reference, and an MIC of 0.84 µg/mL against MRSA clinical). The hybrid **89j** exhibited the best antimicrobial activity against *E. coli* and *P. aeruginosa*, with an MIC of 0.036 and 0.043, respectively, (compared with the reference standard drug ciprofloxacin which has an MIC of 0.056 µg/mL against *E. coli* and an MIC of 1.27 µg/mL against *P. aeruginosa*).

Mohammed et al. [32] synthesized a series of glycosylated-quinoline hybrids derived from fluoroquinolone and studied their antimicrobial properties. The reaction pathway involves the reaction of ciprofloxacin with the corresponding phenolic derivative with an excess of formaldehyde, when the glycosylated-quinoline hybrids **90**–**94** are obtained, Figure 26.

The synthesized glycosylated-quinoline hybrids **90**–**94** were evaluated for their antibacterial activity against various *Gram-positive* and *Gram-negative* bacteria: *Escherichia coli*, *Listeria monocytogenes*, *Salmonella enterica*, *Pseudomonas aeruginosa*, *Listeria monocytogenes*, *E. coli* clinical isolate (resistant to nalidixic acid, ciprofloxacin HCl and norfloxacin antibiotics), methicillin-resistant *Staphylococcus aureus* (MRSA), methicillin-sensitive *Staphylococcus aureus* (MSSA). The hybrids were also tested for their antifungal activity against fungi: *Candida albicans*, *Aspergillus flavus*, *Fusarium solani*, *Stachybotrys chartarum* and *Penicillium chrysogenum*. The hybrid compounds **90**, **91** and **94a** have excellent antimicrobial activity against a fluoroquinolone-resistant *E. coli* clinical isolate, comparable to controls ciprofloxacin and norfloxacin. The hybrid compound **91** also has good antifungal activity against *C. albicans* and *P. chrysogenum*.

Shruthi et al. [33] synthesized a series of piperazine-quinoline hybrids **95a**–**e** and morpholine-quinoline hybrids **96a**–**f** and evaluate them for their antimicrobial properties. The reaction pathway is depicted in Figure 27.

The synthesized hybrids **95a**–**e** and **96a**–**f** were evaluated for their antibacterial (*Acinetobacter baumanii*, *Enterococcus faecium*, *Klebsiella pneumonia*, *Pseudomonas aeruginosa*, *Escherichia coli* and *Staphylococcus aureus*) and antitubercular (*Mycobacterium tuberculosis*) activity. Hybrid **95b** has the best antibacterial activity against *E. coli* and *S. aureus* strains with an MIC of 4, respectively, 2 µg/mL, compared to standard drug vancomycin (MIC of 16, respectively, 0.5 µg/mL). Hybrids **95d**, **95e** and **96f** exhibited the best antibacterial activity against *A. baumaniistains* with an MIC in the range of 1–2 µg/mL, compared to standard drug vancomycin (MIC = 0.5 µg/mL). Hybrids **95b**, **95d** and **95e** also have promising antitubercular activity with an MIC of 4 µg/mL.

Kaur et al. [34] synthesized a series of 3- and 7- substituted-quinoline hybrids derived from fluoroquinolone and studied their antimicrobial properties. The reaction pathway involves the reaction of fluoroquinolone derivatives with the corresponding reagents, when the quinoline hybrids **97**–**104a**,**b** are obtained, Figure 28 and Figure 29.

The synthesized quinoline hybrids **97**–**104a**,**b** were evaluated for their antibacterial activity against four bacterial strains: *Bacillus subtilis*, *Pseudomonas aeruginosa*, *Escherichia coli* and *Staphylococcus aureus*. All hybrids **97**–**104a**,**b** have proved to be active against all bacterial strains, with an MIC value of 25 μg/mL which is fourfold more active compared to the standard drug ciprofloxacin (MIC = 100 μg/mL).

Insuasty et al. [35] synthesized a series of imidazolium-quinoline hybrids and studied their antimicrobial properties. The reaction pathway involves the reaction of 3-formyl-quinolone derivatives with the corresponding imidazolium salts, when the imidazolium-quinoline hybrids **105a**–**h** are obtained, Figure 30.

The synthesized imidazolium-quinoline hybrids **105a**–**h** were evaluated for their antibacterial (*Klebsiella pneumoniae*, *Escherichia coli* and *Staphylococcus aureus*), antifungal (*Cryptococcus neoformans*) and antitubercular (*Mycobacterium tuberculosis H37Rv* and *Mycobacterium bovis BCG*) activities. Hybrid derivatives **105c,d** demonstrated a remarkable antifungal activity against *C. neoformans* (MIC in the range of 15 µg/mL) while for the other fungal strains the activity is weak. The hybrids have modest antibacterial activity (both against *Gram-positive* and *Gram-negative* bacteria) as well as antitubercular activity.

Baartzes et al. [36] synthesized a series of benzimidazole-quinoline and ferrocenyl-quinoline hybrids and studied their antimicrobial properties. The reaction pathway involves the reaction of amino-quinoline derivatives with the corresponding formyl derivatives, when the benzimidazole-quinoline hybrids **106a**–**e** and ferrocenyl-quinoline hybrids **107a**–**e** are obtained, Figure 31.

The synthesized quinoline hybrids **106a**–**e** and **107a**–**e** were evaluated for their antimalarial (*Plasmodium falciparum* and *Plasmodium berghei*) and antitubercular (*Mycobacterium tuberculosis*) activity. All hybrid derivatives are active against tested malaria strains and have modest activity against them. The most active hybrids against malarial strains have proved to be **106c** and **107b**, with an IC_50_ of 0.43, respectively, 0.32 µM, compared with the standard drug chlorquine (IC_50_ = 0.01 µM).

Fedorowicz et al. [37] synthesized a series of zwiterionic hybrids pyridine-fluoroquinolone **108a**–**h** and quinoline-fluoroquinolone **109a**–**h** and studied their antimicrobial properties. The reaction pathway involves a tandem Mannich-electrophilic amination reaction of isoxazolones derivatives and fluoroquinolone bearing a secondary amino group at position 7 of the quinoline ring, Figure 32.

The synthesized quinoline hybrids **108a**–**h** and **109a**–**h** were evaluated for their antibacterial activity against *Gram-positive* and *Gram-negative* bacterial strains (laboratory and clinical: *Staphylococcus aureus ATCC 6538*, *Staphylococcus aureus MRSA N315*, *Staphylococcus epidermidis ATCC 14990*, *Bacillus subtilis ATCC 6633*, *Escherichia coli ATCC 8739*, *Pseudomonas aeruginosa ATCC 9027*, *Proteus vulgaris NCTC 4635*, *Staphylococcus aureus MRSA 6347*, *Staphylococcus epidermidis MRSE 13199* and *Serratia marcescens 12795*) as well as for antibiofilm activity. The hybrid derivatives proved to have bactericidal and antibiofilm activity. The most active hybrids were found to be **109d** and **109e**, exhibiting good inhibition against all strains, with the IC_50_ values in the low micromolar range.

Borazjani et al. [38] synthesized a library of quinoline hybrids (benzothiazole-benzo-quinoline **110**, imino-benzothiazole-benzo-quinoline **111a**–**d**, β-lactam-benzo-thiazole-benzo-quinoline **112a**–**m**) and studied their antimicrobial properties. The reaction pathway involves a [2+2]-cycloaddition reaction of imines **111a**–**d** and ketenes derived from substituted acetic acids, Figure 33.

The synthesized quinoline hybrids **110**–**112** were evaluated for their antimicrobial activity against *Gram-positive* and *Gram-negative* bacterial strains: *Staphylococcus aureus*, *Bacillus subtilis*, *Enterococcus faecalis*, *Salmonella typhi*, *Escherichia coli* and *Pseudomonas aeruginosa*. From the β-lactam class, the assay indicates that the most active hybrids against *E. coli* and *P. aeruginosa*, are **112k** and **112m**, with an MIC of 42, respectively, 20 μg/mL, compared to standard drug gentamycin (MIC of 90, respectively, 5 μg/mL). From the imino-benzothiazole-benzo-quinoline class, the most active hybrids against *P. aeruginosa* and *S. aureus*, are **111a**–**c**, with an MIC of 42 μg/mL, compared to standard drug gentamycin (MIC of 5, respectively, 90 μg/mL).

Berry et al. [39] synthesized a series of peptide-fluoroquinolone hybrids and studied their antimicrobial properties. In order to synthesize the desired hybrids, the authors used solid-phase peptide synthesis, from levofloxacin fluoroquinolone with the corresponding peptide (oligopeptide), when the desired peptide-fluoroquinolone hybrids **113a**–**l** are obtained, Figure 34.

The synthesized peptide-fluoroquinolone hybrids **113a**–**l** were evaluated for their antimicrobial activity against MDR bacterial strains, *Gram-negative* and *Gram-positive*: *Pseudomonas aeruginosa*, *Escherichia coli*, *Klebsiella pneumoniae*, *Acinetobacter baumannii*, methicillin-resistant *Staphylococcus aureus* (MRSA), methicillin-sensitive *Staphylococcus aureus* (MSSA), methicillin-resistant *Staphylococcus epidermis* (MRSE), *Enterococcus faecalis*, *Enterobacter cloacae*, *Stenotrophomonas maltophilia*. The assay indicates that all the peptide-hybrids have weak antibacterial activity. If the hybrids are mixed with fluoroquinolone (ciprofloxacin, levofloxacin and moxifloxacin) drugs, the resulting conjugates possess antimicrobial activity against MDR *Gram-negative* bacteria (clinical isolates, *P. aeruginosa*, *E. coli*, *K. pneumoniae*, *A. baumannii*), superior to reference levofloxacin.

Mermer et al. [40] synthesized a library of triazole- and oxadiazole-fluoroquinolone hybrids and studied their antimicrobial properties. The reaction pathway took placeviaseveral steps of sequential reactions, starting from phenyl piperazine. Finally, the corresponding triazole-fluoroquinolone **114a**,**b** and oxadiazole-fluoroquinolone **115a**–**j** hybrids were obtained *via* a *one-pot* three-component Mannich reaction, Figure 35. The reactions were performed both under conventional thermal heating and microwave, the last pathway being more advantageous.

The synthesized hybrids **114a**,**b** and **115a**–**j** were tested for their antimicrobial activity (against *Gram-positive* and *Gram-negative* strains: *Staphylococcus aureus*, *Enterococcus faecalis*, *Escherichia coli*, *Pseudomonas aeruginosa*, *Klebsiella pneumoniae*, *Acinetobacter haemolyticus*), DNA gyrase and Topoisomerase IV inhibition potentials. The hybrids have good antimicrobial activity and displayed excellent DNA gyrase inhibition. The hybrids **114b**, **115b** and **115h** exhibited the best antimicrobial activity against the tested strains. Thus, the hybrids have excellent activity against *K. pneumoniae* with an MIC of 0.25 µg/mL, compared with the standard drug gentamycin (MIC = 0.25 µg/mL). The hybrids also have excellent activity against *A. haemolyticus* and *P. aeruginosa* with an MIC in the range of 0.5–2 µg/mL, compared with the standard drug gentamycin (MIC = 0.78 µg/mL, respectively, MIC = 1.56 µg/mL). Against *Gram-positive* strain *E. faecalis* the hybrids have excellent activity with an MIC in the range of 0.5–8 µg/mL, compared with the standard drug ampicillin (MIC = 12.5 µg/mL).

Guo et al. [41] synthesized a library of oxadiazole-quinoline hybrids and studied their antibacterial properties. The reaction pathway is straight, involving an alkylation reaction of fluoroquinolone with the corresponding oxadiazole, when the desired oxadiazole-fluoroquinolone hybrids **116a**–**t** were obtained, Figure 36.

The synthesized oxadiazole-fluoroquinolone hybrids **116a**–**t** were tested for their antibacterial activity against methicillin-resistant *Staphylococcus aureus* (MRSA) and laboratory *Staphylococcus aureus*. The hybrids displayed good antibacterial activity, one of the compounds **116k** exhibited excellent antibacterial activity against both methicillin-resistant *S. aureus* and laboratory *S. aureus*, with an MIC in the range of 0.25–2 μg/mL, superior to control drug vancomycin (MIC = 2 μg/mL).

Wang et al. [42] synthesized a series of benzimidazole–quinoline hybrids and studied their antibacterial and antifungal properties. The reaction pathway involves an *N*-alkylation reaction of fluoroquinolone with the corresponding benzimidazole, when the desired benzimidazole-fluoroquinolone hybrids **117a**–**g**, **118a**,**b** and **119a**–**f**, were obtained, Figure 37.

The synthesized benzimidazole-fluoroquinolone hybrids **117a**–**g**, **118a**,**b** and **119a**–**f** were screened against *Gram-positive* and *Gram-negative* bacteria, respectively, fungus (methicillin-resistant *Staphylococcus aureus* (MRSA), *Enterococcus faecalis*, *Staphylococcus aureus*, *Staphylococcus aureus ATCC25923*, *Staphylococcus aureus ATCC29213*, *Klebsiella pneumonia*, *Escherichia coli*, *Pseudomonas aeruginosa*, *Acinetobacter baumanii*, *Pseudomonas aeruginosa ATCC27853*, *Escherichia coli ATCC25922*, *Candida albicans*, *Candida tropicalis*, *Aspergillus fumigatus*, *Candida albicans ATCC90023*, *Candida parapsilosis ATCC22019*). The results of the assay were promising, with some hybrids having excellent antibacterial activity. The most active hybrids against *K. pneumonia* are **117a** and **117c**, with an MIC of 8 μg/mL, compared to the standard drug norfloxacin (MIC > 512 μg/mL).The most active hybrids against *S. aureus* are **119a** and **119f**, with an MIC of 4 μg/mL, compared to the standard drug norfloxacin (MIC = 64 μg/mL).

Bharadwaj et al. [43] synthesized a series of oxadiazole–quinoline hybrids and studied their antibacterial and antifungal properties. The reaction pathway involves a cyclocondensation reaction of hydrazinyl-quinoline derivative with the corresponding aromatic acids, when the desired oxadiazole–quinoline hybrids **120a**–**g** were obtained, Figure 38.

The synthesized oxadiazole–quinoline hybrids **120a**–**g** were tested against clinical isolates *Gram-positive* and *Gram-negative* bacteria (*Staphylococcus aureus*, *Bacillus cereus*, *Escherichia coli*, *Serratia marcescens*), respectively, fungus (*Aspergillus niger*, *Trichophyton mentagrophytes*, *Candida albicans*, *Candida parapsilosis*). The antimicrobial activity of oxadiazole–quinoline derivatives was good, the hybrids **120a** and **120f** having the best antimicrobial activity against *B. cereus* with an MIC of 17, respectively, 24 μg/mL, compared to standard drug ampicilin (MIC = 16 μg/mL).

Tahaab et al. [44] synthesized a series of oxadiazole–quinoline hybrids and studied their leishmanicidal potential. The reaction pathway to obtain the oxadiazole–quinoline hybrids **121a**–**r** is depicted in Figure 39.

The synthesized oxadiazole-quinoline hybrids **121a**–**r** were tested for their leishmanicidal activity against *Leishmania major* promastigote. Most of the synthesized hybrids have a good leishmanicidal activity, compound **121r** was found to be the most active (IC_50_ = 0.10 μM) from the series, being 70 times more active than the standard drug (pentamidine, IC_50_ = 7 μM).

Irfan et al. [45] synthesized a series of triazole–quinoline hybrids and studied their antifungal properties. The reaction pathway involves a typical click cyclocondensation reaction of azide with a compound with a triple bond, when the desired triazole–quinoline hybrids **122a**–**c** were obtained, Figure 40.

The synthesized triazole–quinoline hybrids **122a**–**c** were tested against fungus *Candida albicans*, both clinical isolates and laboratory strains [three FLC susceptible strains (*C. albicans D27*, *C. albicans D31* and *C. albicans D39*) and one FLC resistant strain (*C. albicans D15.9*)]. The best antifungal activity was found for the hybrids **122a** and **122b**, having an MIC of 25 μg/mL for **122a** and an MIC of 250 μg/mL for **122b**, compared to control FLC (MIC >1 μg/mL)

Pandya et al. [46] synthesized a library of pyrazole–isoquinoline hybrids and studied their antimicrobial properties. The reaction pathway involves a palladium-catalyzed reaction of pyrazole derivatives with *t*-butyl-isocyanide, when the corresponding pyrazole–isoquinoline hybrids **123a**–**g**, were obtained, Figure 41.

The synthesized pyrazole–isoquinoline hybrids **123a**–**g** were evaluated for their antimicrobial activity against different pathogenic strains: bacterial strains (*Staphylococcus aureus*, *Escherichia coli*, *Enterococcus faecalis*, *Streptococcus pyogens* and *Vibrio cholera*), fungal strains (*Candida albicans*, *Candida glabrata*, *Candida krusei*, *Candida tropicalis* and *Candida parapsilosis*) and tubercular strain (*Mycobacterium tuberculosis*). The antimicrobial activity of hybrids was very good, the hybrids **123e** and **123g** having the best antimicrobial activity, compared to standard drugs kanamycin and amphotericin B. Thus, the most active hybrids against *S. aureus* are **123e** and **123g**, having an MIC of 20 μM, respectively, 37 μM, compared to standard drug kanamycin (MIC of 31 μM). The most active hybrids against *V. cholera* are **123e** and **123g**, having an MIC of 41 μM, respectively, 90 μM, compared to the standard drug kanamycin (MIC of 62 μM). The hybrids **123e** and **123g** have the best antitubercular activity against *M. tuberculosis* with an MIC of 30 μg/mL, respectively, 32 μg/mL, compared to standard drugs rifampicin and isoniazide (MIC of 90 μg/mL).

Verma et al. [47] obtained a series of piperazine- and pyrimidine- isoquinoline hybrids and studied their antimicrobial properties. The piperazine-isoquinoline hybrids **126a**–**h** were synthesized by condensation of the carboxylic acid intermediates **124a**–**d** with appropriate aryl-piperazines, Figure 38. The pyrimidine-isoquinoline hybrids **127a**–**h** were synthesized in two steps: an *O*-alkylation of the carboxylic acid intermediates **124a**–**d** (with ethylene dichloride), followed by an *S*-alkylation of the obtained compounds **125a**–**d** (with thio-pyrimidine), Figure 42.

The synthesized piperazine- and pyrimidine-isoquinoline hybrids **124a**–**h** and **125a**–**h** were evaluated for their antibacterial and antifungal (*Escherichia coli*, *Klebsiella pneumoniae*, *Staphylococcus aureus*, *Bacillus subtilis*, *Aspergillus niger*, *Aspergillus oryzae*, *Candida albicans* and *Pencillium chrysogenum*), antioxidant, anticancer and antituberculosis (*Mycobacterium tuberculosis*) activities. The antibacterial assay indicates that three hybrids, namely **124a**, **125a** and **126e** have the best activity against *E. coli* (with an MIC in the range of 1–3 μg/mL) and *K. pneumoniae* (with an MIC in the range of 1.5–3 μg/mL), compared with the standard drug ciprofloxacin (MIC = 1.5 μg/mL). The hybrids **125a**, **126a** and **127a** also have excellentactivity against *S. aureus* (with an MIC in the range of 1–3 μg/mL) and *B. subtilis* (with an MIC in the range of 1.5–3 μg/mL), compared with the standard drug ciprofloxacin (MIC = 1.5 μg/mL, respectively, MIC = 3 μg/mL). The hybrids **125a**, **126a** and **127a** have excellent activity against fungus *A. niger*, *C. albicans*, *A. oryzae*, and *P. chrysogenum* (with an MIC of 1.5 μg/mL), compared with the standard drug fluconazole (MIC = 1.5 μg/mL for *A. niger* and *C. albicans*, respectively, MIC = 3 μg/mL for *A. oryzae*, and *P. chrysogenum*). The hybrids **127b** and **127e** have the best activity against *M. tuberculosis* (MIC 1.0 mg/mL), compared with the standard drug rifampicin (MIC = 0.1mg/mL. The antioxidant and anticancer activity proved to be modest.

### 2.3. Our Group Recent Contributions

Our concern for obtaining new six-member ring azaheterocycle entities with antimicrobial activity for medicinal chemistry applications was started three decades ago [48,49,50,51] when we tried to obtain new diazines with good to excellent antibacterial and antifungal activities. Further, we will present some recent results obtained by us in the field of hybrid azines with antimicrobial activity.

In continuation of our concern for new compounds with antimicrobial activity, Diaconu et al. [52] synthesized a large library of hybrid imidazole- and benzimidazole-quinoline derivatives and studied their antimicrobial properties. The reaction pathway (Figure 42) involves an initial *N*-acylation reaction of 8-aminoquinoline, followed by an *N*-alkylation of the -NH- amino group from imidazole/benzimidazole heterocycle, when the key imidazole-quinoline **128a**,**b** and benzimidazole-quinoline **129a**,**b** hybrids are obtained. Next, a quaternization reaction of *N*-imidazole atom with activated halogenated compounds leads to a second class of hybrids, the quaternary salts of imidazole- benzimidazole-quinolines, **130a**–**k** and **132a**–**k** (with one methylene group as linker) and **131a**–**k** and **133a**–**k** (with two methylene groups as linker). Finally, imidazolium and benzimidazolium ylides (generated in situ from the corresponding salts) react with dimethyl acetylenedicarboxylate (DMAD), generating another class of hybrid quinoline derivatives, the benzimidazole-quinoline cycloadducts **134a**–**k**, Figure 43.

The synthesized hybrids were evaluated for their antimicrobial (*Staphylococcus aureus*, *Escherichia coli* and *Candida albicans)* and anticancer activities. The results of the antibacterial assay indicate that some hybrid compounds are biologically active in the range of nano-molar, five benzimidazole-quinoline hybrid salts (**133c**, **133d**, **133f**, **133h**, **132h**) have excellent activity against *Gram-negative* bacteria *E. coli* (DIZ in the range of 20–24 mm) superior to control gentamicin (DIZ of 12 mm) and one compound (**131i**) have excellent activity against *Gram-positive* bacteria *S. aureus* (DIZ of 20 mm) superior to control gentamicin (DIZ of 14 mm). The anticancer assay indicates that some benzimidazole-quinoline hybrid salts (**133h**, **132h**, **133c**, **133f**) have excellent anticancer activity in the range of nano-molar, against some cancer cells (Leukemia, Breast cancer, Lung cancer and Ovarian cancer). Interesting SAR correlations have been performed.

In another research work, Diaconu et al. [53] synthesized a new series of hybrid imidazole- and benzimidazole-pyridine derivatives and studied their antimicrobial properties. The reaction pathway (Figure 43) involves an *N*-acylation reaction of 2-aminopyridine, followed by an *N*-alkylation of the -NH- amino group from imidazole/benzimidazole heterocycle when the corresponding imidazole-pyridine **135** and benzimidazole-pyridine **136** hybrids are obtained. Finally, a quaternization reaction of *N*-imidazole atom with activated halogenated compounds leads to a second class of hybrids, the imidazole-pyridine **137a**,**b** and benzimidazole-pyridine **138a**,**b** salts, Figure 44.

The synthesized hybrids were tested for their antimicrobial activities against *Staphylococcus aureus*, *Escherichia coli* and *Candida albicans*. The results of the antibacterial assay indicate that the imidazole- and benzimidazole-pyridine hybrids have interesting antimicrobial properties, especially the hybrid benzimidazole-pyridine salt **138a** have a powerful antibacterial activity against *Gram-positive* strain *S. aureus* and *Gram-negative* germ *E. coli* (DIZ of 30 mm), superior to control drug gentamicin (DIZ of 18 mm).

Antoci et al. [54] synthesized a new series of hybrid *bis*(imidazole)- and *bis*(benzimidazole)-pyridine derivatives and studied their antimycobacterial activity. The reaction pathway (Figure 44) involves an *N*-alkylation of the -NH- amino group from imidazole/benzimidazole heterocycle when the corresponding *bis*(imidazole)-pyridine **139, 140** and *bis*(benzimidazole)-pyridine **141** hybrids are obtained. In the next step, a quaternization reaction of *N*-imidazole atom with activated halogenated compounds leads to a second class of hybrids, the *bis*(imidazole)-pyridine **142a**–**g** and *bis*(benzimidazole)-pyridine **143a**–**g** salts, Figure 45.

The synthesized hybrids were tested in a primary screening for their antimycobacterial activities against *M. tuberculosis H37Rv* under aerobic conditions, eight hybrids (namely **140**, **141**, **142e**,**f** and **143c**,**e**–**g**) having excellent activity against *Mtb H37Rv*, with an MIC in the range of 17–92 μM. The most active antimycobacterial five compounds (namely **140**, **141**, **142f** and **143c**,**f**) were subjected to the secondary antimycobacterial assay. The obtained results indicate that our compounds are very active against both replicating and non-replicating *Mtb* (superior to control metronidazole), exhibited excellent intracellular activity, are active against drug-resistant *Mtb* strains, have no cytotoxicity, and three of them (**142f** and **143c**,**f**) have a bactericidal mechanism of action. The results of ADMET pharmacokinetic experimental studies for hybrid **143f**, reveal that this compound is truly a candidate for a future drug: a lower clearance rate, a great half-time in vivo, a low potential for drug–drug interactions with a high duration of action and lack of cytotoxicity. The best antitubercular activity has the hybrid **143e** with an MIC of 17 μM, MBC of 50 μM, IC_50_ of 9 μM. Under anaerobic conditions (LORA) the hybrid **143e** have the MIC of 120 μM and IC_50_ of 9 μM. Against resistant isolates of *Mtb* strains [five strains, INH-R1 and INH-R2 (strains resistant to isoniazid), RIF-R1 and RIF-R2 (strains resistant to rifampicin), FQ-R1 (strain resistant to fluoroquinolone)] the hybrid **143e** have the MIC in the range of 10–30 μM and IC_50_ in the range of 10–20 μM. Against nontuberculous mycobacteria *Mycobacterium avium* and *Mycobacterium abscessus*, the hybrid **143e** has the MIC in the range of 50–80 μM and IC_50_ in the range of 30–50 μM. The intracellular activity and cytotoxicity of the hybrid **143e** were IC_50_ of 14 μM, respectively, and IC_50_ of 50 μM.

Furthermore, in a subsequent paper [55], some of the above authors performed a thorough molecular docking study in order to determine the binding sites and ADMET properties of the hybrid *bis*(imidazole)- and *bis*(benzimidazole)-pyridine derivatives. The obtained results indicate the most probable binding sites the G-quadruplex DNA string and DNA strain in complex with dioxygenase. The predicted ADMET properties are in accordance with the experimental one presented above [54].

Continuing our studies in the field of hybrid pyridine and quinoline derivatives, Mantu et al. [56] synthesized a new series of hybrid imidazole- and benzimidazole-pyridine and quinoline derivatives and studied their antimicrobial properties. The reaction pathway involves a quaternization reaction of *N*-imidazole atom with activated halogenated compounds, when the corresponding salts of imidazole- and benzimidazole-pyridine and quinoline hybrids are obtained, Figure 46.

The synthesized hybrids were tested for their antimycobacterial and anticancer activities. The antimycobacterial assay reveals that our hybrids have modest activity against *Mtb* strains. The anticancer assay indicates that one of the hybrids, namely **129a**, has a very good and selective antitumor activity against Renal Cancer A498 and Breast Cancer MDA-MB-468.

Diaconu et al. [57] synthesized two new series of hybrid quinoline-sulfonamide complexes and studied their antimicrobial activity. The reaction pathway involves a straight and efficient two-step procedure. In the first step, an acylation reaction of (3-, 4- or 8-)aminoquinoline derivatives with the corresponding benzenesulfonyl chlorides **148a**–**c** or quinolylsulfonyl chloride **148d** took place, the desired ligands quinoline-sulfonamide type **149a**–**d** being obtained. In the second step, a complexation reaction of ligands **149a**–**d** with metal acetate (Cu^2+^, Co^2+^, Cd^2+^) or chloride (Zn^2+^) took place, with the desired hybrids quinoline-benzene-sulfonamide complexes (**150a**–**d**, **151a**–**d** and **152a**–**d**) and quinoline-quinolinyl-sulfonamide complexes **153a**–**d** being obtained. The reaction pathway is depicted in Figure 47 for the complexes derived from 8-aminoquinoline.

The synthesized hybrids were tested for their antimicrobial activity, with some of them having a very good antibacterial (*Staphylococcus aureus*, *Escherichia coli*) and antifungal (*Candida albicans*) activity. For instance, the hybrid *N*-(quinolin-8-yl)-4-chloro-benzenesulfonamide cadmium **153d** has the best antibacterial activity, with a DIZ of 21 mm and an MIC of 19.04 × 10^−5^ mg/mL against *S. aureus*, a DIZ of 19 mm and an MIC of 609 × 10^−5^ mg/mL against *E. coli*, and an excellent antifungal activity against *C. albicans*, with a DIZ of 25 mm and an MIC of 19.04 × 10^−5^ mg/mL.

Al-Matarneh et al. [58] synthesized two new series of pyrrolo-quinoline and pyrrolo-isoquinoline hybrids and studied their antimicrobial activity. The reaction pathway involves a 3 + 2 dipolar cycloaddition reaction of the quinolinium and isoquinolinium ylides (generated in situ from the corresponding salts) with *N*-ethyl- or *N*-phenyl-maleimide, when the hybrid spyrrolo-quinoline **154a**–**c** and pyrrolo-isoquinoline **155a**–**c** are obtained, Figure 48.

The synthesized hybrids pyrrolo-quinoline **154a**–**c** and pyrrolo-isoquinoline **155a**–**c** were tested for their antimicrobial activities but, unfortunately, the hybrids have no significant activity.

Danac et al. [59] synthesized a series of pyrrolo-phenanthroline hybrids and studied their antimycobacterial activity. The reaction pathway involves a 3 + 2 dipolar cycloaddition reaction of the phenanthrolinium ylides (generated in situ from the corresponding salts) with *N*-ethyl- or *N*-phenyl-maleimide, when the pyrrolo-phenanthroline hybrids **156a**–**c** are obtained, Figure 49.

The synthesized hybrids **156a**–**c** were tested for their antimycobacterial activities. The antimycobacterial assay reveals that one hybrid, **156a**, has a strong activity against the *Mtb* strain, with an IC_50_ of 56 μM.

Danac, Olaru et al. [60,61] synthesized a library of indolizine-pyridine hybrids and studied their antimycobacterial activity. The reaction pathway involves a 3 + 2 dipolar cycloaddition reaction of the 4,4′-bipyridinium mono-ylides (generated in situ from the corresponding salts) with ethyl propiolate, when the mono-indolizine-pyridine hybrids **157a**–**e** are obtained, Figure 49. Next, a quaternization reaction of pyridine nitrogen atom with activated halogenated compounds **158a**–**l** is leading to the salts of mono-indolizine-pyridine hybrids **159a**–**l**. Finally, another 3 + 2 dipolar cycloaddition reaction with ethyl propiolate is leading to *bis*-indolizine-pyridine hybrids **160a**–**d**, Figure 50.

The synthesized hybrids were tested in a primary screening for their antimycobacterial activities against *M. tuberculosis H37Rv* under aerobic conditions, the salts of mono-indolizine-pyridine hybrids **159a**–**l** displaying an excellent activity against *Mtb H37Rv*, superior to the second-line antitubercular drugs cycloserine and pyrimethamine and, equal as the first line anti-TB Ethambutol. The most active antimycobacterial five compounds (namely **159a**, **159c**, **159d**, **159h** and **159i**) were subjected to the secondary antimycobacterial assay (MIC, MBC, LORA, intracellular (macrophage) drug screening, and MTT cell proliferation). These mono-indolizine-pyridine hybrids have proved to be very active against replicating and non-replicating *M. tuberculosis*, are active against both extracellular and intracellular organisms, have a bactericidal mechanism of action, and had basically no toxicity. The best antitubercular activity has the hybrid **159i** with an MIC of 8 μM, MBC of 3 μM, IC_50_ of 7 μM. Under anaerobic conditions (LORA) the hybrid **159i** have the MIC of 63 μM and IC_50_ of 1.9 μM. Against resistant isolates of *Mtb* strains [five strains, INH-R1 and INH-R2 (strains resistant to isoniazid), RIF-R1 and RIF-R2 (strains resistant to rifampicin), FQ-R1 (strain resistant to fluoroquinolone)] the hybrid **159i** have the MIC in the range of 6–22 μM and IC_50_ in the range of 6–12 μM. Against nontuberculous mycobacteria *Mycobacterium avium* and *Mycobacterium abscessus*, the hybrid **159i** has the MIC in the range of 23–50 μM and IC_50_ in the range of 14–18 μM. The intracellular activity and cytotoxicity of the hybrid **159i** were IC_50_ of 5 μM, respectively IC_50_ of 2 μM.

## 3. Perspectives and Conclusions

To conclude, we report herein the latest recent advances concerning the synthesis and antimicrobial properties of hybrid azine derivatives. The literature data presented in this review indicate that there is a great urgency insociety and pharmaceutical industry to develop new antimicrobial drugs for the treatment of infectious diseases. Moreover, the data indicate that a modern approach used to overcome the drawbacks of infectious diseases is to use molecular hybridization strategy, as a new modern approach in drug discovery. The hybrid pyridine, quinoline, isoquinoline and their fused derivatives have invaluable importance in modern antimicrobial therapy, the results presented in this review indicate that they have a large variety of antimicrobial activities, including antibacterial, antifungal, antimycobacterial, antileishmanial, antimalarial, antiviral, etc.

We show that the best methods for the synthesis of hybrid compounds are cyclocondensation, condensation and simple typical organic chemistry reactions such as alkylation, acylation, etc. We also show that many hybrid compounds have excellent antimicrobial activity, the combination of an azine moiety with a five-member ring azaheterocycle being the best approach to obtain drugs with improved and superior antimicrobial properties. A special mention has to be made about the obtained results in antituberculosis therapy, where the use of hybrids with pyridine or by pyridine merged with an imidazole or benzimidazole moiety seems to be a very efficient approach in treatment, in some cases, the hybrids had a spectacular antitubercular activity, including a bactericidal mechanism of action. Moreover, the fact that some of these hybrids are in different clinical trials is a good and solid argument for further research in this field.

Finally, having in view the above consideration, we encourage and underline further studies in the field of hybrid azine merged with a five-member ring azaheterocycle, which appears to be the most promising field of research within this area.

## Data Availability

Not applicable.

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
