# Peer review of "Hybrid Azine Derivatives: A Useful Approach for Antimicrobial Therapy"

_pharmaceutics, 2022, doi:10.3390/pharmaceutics14102026_

Round 1

Reviewer 1 Report

Overall the manuscript is technically sound. The revision performed is well organized and a significant contribution to the field.

Main comment: 

To simplify the article's content, the authors must include a table as a summary with the following three sections: 1) synthesized hybrid derivatives, 2) type of reaction (used in the synthesis), and 3) biological activity (divided into three subsections: antimicrobial activity/strains used/effect observed.

Minor comments:

1. "via minimal inhibitory concentration (MIC) method" is repeated on page 2, line 57, and page 2, line 77/page 3, line 78.

2. Pag 6, line 156: "vitro" must be italicized.

3. Reference 8 is missed.

Author Response

Thank you very much for all valuable comments.

  • To simplify the article's content, the authors must include a table as a summary with the following three sections: 1) synthesized hybrid derivatives, 2) type of reaction (used in the synthesis), and 3) biological activity (divided into three subsections: antimicrobial activity/strains used/effect observed.

Response.       

We done it, on pages 3-6.

Minor comments:

  • "via minimal inhibitory concentration (MIC) method" is repeated on page 2, line 57, and page 2, line 77/page 3, line 78.

We made the changes.

  • Pag 6, line 156: "vitro" must be italicized.

It has been done all around the text.

  • Reference 8 is missed.

We introduced.

Reviewer 2 Report

My previous comments (not all)  justify the reviewer 3 comments__

The introduction is according to the aim of this research, but it lacks potential pharmacological applications or reported patents analyses related to this topic due to the aim of the authors to include the antimicrobial therapy concept.

- I suggest selecting the more active molecules for each scheme throughout the manuscript due to all the information was reported previously by other authors (highlighted in yellow). According to this possible selection, all the schemes will only show the most important representatives’ compounds.

Author Response

Reviewer 2 and 3

Thank you very much for all valuable comments.

The present review manuscript entitled "Hybrid azine derivatives: an useful approach for antimicrobial therapy" from Amariucai-Mantu et al involved the synthesis and antimicrobial activity analyses of hybrid azine derivatives with potential drug applications.

- The introduction is according to the aim of this research, but it lacks potential pharmacological applications or reported patents analyses related to this topic due to the aim of the authors to include the antimicrobial therapy concept. Moreover, the authors had achieved a complete bibliographical evaluation of current produced molecules with potential antimicrobial activity against several Gram+ and Gram- strains.

Response.

We introduced into the text the pharmacological applications of compounds,

  • on pages 1, rows 40-43 and Scheme II:

“ As a result of this approach, important advances have been done in the antimicrobial therapy, some of the present drugs from the market having a hybrid structure (Scheme I) and some hybrid structures are in different clinical trials (Scheme II) [2-8].”

  • on pages 2, rows 49-52, 56-59 and Scheme I:

“As a matter of fact, the greatest majority of the existing drugs from the market contain in their structure a nitrogen heterocycle, some of them being with hybrid structure (Scheme I), which justify the demand of pharmaceutical industry for such of drugs with nitrogen heterocycle skeleton.

Because of the above considerations, there is a big and urgent demand from pharmaceutical industry for newer and better drugs with enhanced antimicrobial activity, with superior pharmacokinetic and pharmacodynamic properties, the hybrid drugs being a serious and preferential option.”

Besides, I encourage the authors to check some mistakes such as (highlighted in yellow, pdf attached):

  • Title: the correct spelling is to use “a useful”

We corrected.

  • Please remember to use italics for R when nomenclature is using

It has been done all around the text.

- Please homologate the size and the style of the schemes throughout the manuscript. Also, they look blurry even at 100%-page size. It should be important to check the object setting properties according to the software the authors used.

The size and the style of the schemes, as well as the resolution have been improved.

- Please, take care about the length of the bonds between atoms (e.g.: scheme 3, highlighted in yellow)- Scheme 17. Both arrows should be at the same level in the graphics.

It has been done all around the text.

- I suggest selecting the more active molecules for each scheme throughout the manuscript due to all the information was reported previously by other authors (highlighted in yellow). According to this possible selection, all the schemes will only show the most important representatives’ compounds.

We indicate into the text for each series of compounds, which one is the most active, with the corresponding MIC, DIZ or IC50 values (highlighted with yellow). Into the reaction schemes we preferred to keep all compounds.

  • Although, I suggest improving the conclusions and future perspectives mentioning (venturing thoughts) which will be the possible clinical formulation/applications improvements that these kind of hybrid azine derivatives compounds will need (considering the Pharmaceutics scope). Additionally, taking into account the focus of this review is the antimicrobial activity of these compounds the authors should emphasize this important property for selected compounds in the concluding remarks item. (Some questions that help to explain this point could be: Which are the most promising compounds and why, from a critical point of view. Currently some of them are included in clinical trials assays?).

Response

We completely rewrite the conclusions part according with the referee suggestions, on pag 35-36, rows, 945-970:

“To conclude, we report herein the latest recent advances concerning the synthesis and antimicrobial properties of hybrid azine derivatives. The literature data presented in this review indicate that there is a great urgency from the society and pharmaceutical industry do develop new antimicrobial drugs for the treatments of infectious diseases. Also, the data indicate that a modern approach used to overcome the drawbacks of infectious diseases is to use molecular hybridization strategy, as a new modern approach in drug discovery. The hybrid pyridine, quinoline, isoquinoline and their fused derivatives have an invaluable importance in modern antimicrobial therapy, the results presented in this review indicating that they have a large variety of antimicrobial activities, these including antibacterial, antifungal, antimycobacterial, antileishmanial, antimalarial, aniviral, etc.

We show that the best ways for synthesis of hybrid compounds are cyclocondensation, condensation and simple typical organic chemistry reactions such alkylation, acylation, etc. We also show that many hybrid compounds have excellent antimicrobial activity, the combination of an azine moiety with a five member ring azaheterocycle being the best approach to obtain drugs with improved and superior antimicrobial properties. A special mention has to be pointed out to the obtained results in antituberculosis therapy, were the using of hybrids with pyridine or bypyridine merged with an imidazole or benzimidazole moiety seems to be a very efficient approach in treatment, in some cases the hybrids having a spectacular antitubercular activity, these including and a bactericidal mechanism of action. Moreover, the fact that some of these hybrids are in different clinical trials is a good and solid argue for further research in this field.

Finally, having in view the above consideration, we encourage and underline further studies in the field of hybrid azine merged with a five member ring azaheterocycle, which appear to be the most promising field of research within this area’s.”

  • Finally, I would like to invite the authors to include the abbreviation list of words at the end of this manuscript.

We done it, at the end of manuscript, pag 36-37.

Reviewer 3 Report

The present review manuscript entitled "Hybrid azine derivatives: an useful approach for antimicrobial therapy" from Amariucai-Mantu et al involved the synthesis and antimicrobial activity analyses of hybrid azine derivatives with potential drug applications.

The introduction is according to the aim of this research, but it lacks potential pharmacological applications or reported patents analyses related to this topic due to the aim of the authors to include the antimicrobial therapy concept.

Moreover, the authors had achieved a complete bibliographical evaluation of current produced molecules with potential antimicrobial activity against several Gram+ and Gram- strains.

Besides, I encourage the authors to check some mistakes such as (highlighted in yellow, pdf attached):

- Title: the correct spelling is to use “a useful”

- Please remember to use italics for R when nomenclature is using

- Please homologate the size and the style of the schemes throughout the manuscript. Also, they look blurry even at 100%-page size. It should be important to check the object setting properties according to the software the authors used.

- Please, take care about the length of the bonds between atoms (e.g.: scheme 3, highlighted in yellow)

- Scheme 17. Both arrows should be at the same level in the graphics.

- I suggest selecting the more active molecules for each scheme throughout the manuscript due to all the information was reported previously by other authors (highlighted in yellow). According to this possible selection, all the schemes will only show the most important representatives’ compounds.

Although, I suggest improving the conclusions and future perspectives mentioning (venturing thoughts) which will be the possible clinical formulation/applications improvements that these kind of hybrid azine derivatives compounds will need (considering the Pharmaceutics scope). Additionally, taking into account the focus of this review is the antimicrobial activity of these compounds the authors should emphasize this important property for selected compounds in the concluding remarks item. (Some questions that help to explain this point could be: Which are the most promising compounds and why, from a critical point of view. Currently some of them are included in clinical trials assays?).

Finally, I would like to invite the authors to include the abbreviation list of words at the end of this manuscript.

I recommend the acceptance of this manuscript after the authors performed the suggested corrections/additions.

Author Response

(The authors gave the same response as above.)

Round 2

Reviewer 2 Report

The revision is satisfactory, and authors rectified the sections as I quoted earlier. Hence I recommended for publication. 

Reviewer 3 Report

The authors preformed all the suggested corrections/additions.